# A Single Arm Clinical Study on the Effects of Continuous Erythropoietin Receptor Activator Treatment in Non-Dialysis Patients with Chronic Heart Failure and Renal Anemia

**DOI:** 10.3390/biomedicines11030946

**Published:** 2023-03-20

**Authors:** Akira Sezai, Hisakuni Sekino, Makoto Taoka, Shunji Osaka, Masashi Tanaka

**Affiliations:** 1Department of Cardiovascular Surgery, Nihon University School of Medicine, Tokyo 173-8610, Japan; 2Sekino Hospital, Tokyo 171-0014, Japan

**Keywords:** renal anemia, heart failure, erythropoietin

## Abstract

Erythropoiesis-stimulating agents improve the NYHA functional class and decrease the hospital readmission rates for heart failure; however, little is known about the influence of continuous erythropoietin receptor activator (CERA) on the heart. Therefore, a prospective study was conducted to investigate the effects of CERA on cardiac and renal function and oxidative stress in chronic heart failure with renal anemia. Sixty patients with chronic heart failure and renal anemia were enrolled and received CERA for 12 months. The primary endpoints were hemoglobin (Hb) and hematocrit, and the secondary endpoints were: (1) atrial natriuretic peptide (ANP) and B-type natriuretic peptide (BNP); (2) NYHA class; (3) echocardiography; (4) blood urea nitrogen, creatinine, cystatin C, and urinary albumin; (5) high-sensitivity C-reactive protein; (6) oxidized low-density lipoprotein (Ox-LDL); and (7) renin, angiotensin-II, and aldosterone. There was a significant difference in the Hb levels measured before and after CERA administration. The BNP, ANP, NYHA, left ventricular mass index, renal function, and Ox-LDL decreased significantly after CERA administration. This study shows that CERA improves anemia and reduces renal impairment, as well as cardiac and oxidative stress. The result of this study is useful for a study in which switching from CERA to a new renal anemia drug, hypoxia-inducible factor prolyl-hydroxylase inhibitor, is investigated.

## 1. Introduction

Chronic kidney disease (CKD) and anemia are independent prognostic factors for heart failure and are associated with poor prognosis [1,2,3]. Furthermore, patients with CKD may develop cardio-renal-anemia syndrome, a vicious cycle involving heart disease, renal disease, and anemia. This syndrome leads to the exacerbation of both heart and renal disease because the associated ischemia causes fluid retention and inflammation and anemia causes oxidative stress [1,4]. Furthermore, the presence of anemia is considered to negatively affect the efficacy of heart failure drugs [5]. The treatment of anemia is also important in heart failure management. In the CHART-2 (Chronic Heart Failure Analysis and Registry in the Tohoku District-2) study performed in Japan, 35% of patients with chronic cardiac failure had anemia [6]. The underlying mechanism of anemia in patients with heart failure are as follows: (1) Dilution by retained body fluid; and (2) decreased hematopoiesis. In the mechanism of (1), the renal blood flow is decreased, secondary to decreased cardiac output, thereby renin-angiotensin-aldosterone is activated. This results in the retention of body fluid. (2) is the decreased production of erythropoietin or decreased response by bone marrow cells to erythropoietin secondary to CKD complication [7]. Other causes of anemia include iron deficiency, decreased utilization of retained iron due to inflammation [5], high prevalence of malignancy in elderly patients with heart failure, use of anticoagulant/anti-platelet in patients with heart failure, and digestive tract bleeding. As the cause of anemia in heart failure is unknown, there is currently no established treatment. However, studies are currently underway to evaluate the effects of erythropoiesis-stimulating agents (ESA) in patients with anemia and heart failure. ESA were shown to improve the New York Heart Association (NYHA) functional class, decrease the hospital readmission rates for heart failure, and increase exercise tolerance [8,9,10]. No studies to date have evaluated whether ESA improve all-cause mortality, exacerbate heart failure, or improve hospital readmission rates, but the large-scale RED-HF (Reduction of Events by Darbepoetin Alfa in Heart Failure) study found that they can cause thromboembolism [11].

Among the many studies on the effects of ESA in renal anemia, one reported that continuous erythropoietin receptor activator (CERA; epoetin beta pegol, Mircera^®^, Chugai Pharmaceutical Co., Ltd., Tokyo, Japan) has a long half-life and good efficacy [12], and its clinical benefits have been reported [13,14]. However, there is no clinical study report of CERA in patients with heart failure. Accordingly, we believed this study was warranted. In clinical practice, it is found that the improvement of anemia sometimes prevents the worsening of heart failure. Therefore, a prospective clinical study was conducted to investigate the influence of CERA on cardiac function, renal function, and oxidative stress in patients with chronic heart failure and renal anemia.

## 2. Materials and Methods

### 2.1. Study Design

This was a single arm design at a single clinical site.

Subjects: Participants were patients with chronic heart failure and renal anemia being treated at Sekino Hospital, a logistical support hospital of the Nihon University Itabashi Hospital. The definition of chronic heart failure in the study population in this study is as follows: Those classified as NYHA I to III, receiving follow-up at an out-patient clinic who experienced acute decompensated heart failure, not re-admitted due to worsening of heart failure for at least past one year and receiving standard heart failure drugs (diuretics, β-blockers, and renin-angiotensin system inhibitors).

Renal anemia was defined as hemoglobin (Hb) levels of 11 g/dL or less; CKD stage 3b or lower; negative fecal occult blood test results; anemia that is not iron deficiency anemia, pernicious anemia, or hemolytic anemia; no vitamin B12 or folic acid deficiency; erythropoietin levels of 50 U/mL or less; and those whose anemia was not improved by iron treatment after the diagnosis of iron deficiency anemia.

Methods: CERA was administered intravenously as an add-on treatment once a month or once every two weeks for 12 months. The initial dose of CERA was 50 µg/month, and the dose was then adjusted to less than 13 g/dL in order to achieve Hb levels of 11. If the patient’s Hb level was 13 g/dL or higher, the administration of CERA was either stopped or reduced.

Endpoints: The primary endpoints were changes in the levels of Hb and hematocrit (Ht), and the secondary endpoints were changes in the following parameters: (1) atrial natriuretic peptide (ANP) and B-type natriuretic peptide (BNP); (2) NYHA class; (3) echocardiography measurements of left ventricular ejection (LVEF), fractional shortening (%FS), left ventricular end-diastolic dimension (LVEDD), left ventricular end-systolic dimension (LVESD), left ventricular mass index (LVMI), and E/e’ ratio; (4) blood urea nitrogen (BUN), serum creatinine (sCr), estimated glomerular filtration rate (eGFR), cystatin C, and urinary albumin excretion adjusted for urinary albumin (U-Alb); (5) high-sensitivity C-reactive protein (hs-CRP); (6) oxidized low-density lipoprotein (Ox-LDL); and (7) renin, angiotensin-II, and aldosterone.

The Hb, Ht, ANP, BNP, sCr, eGFR, cystatin C, U-Alb, hs-CRP, and Ox-LDL levels were measured and the NYHA class was assessed at baseline before CERA administration and after 1, 3, 6, and 12 months of treatment. At baseline and after 6 and 12 months of treatment, the levels of renin, angiotensin-II, and aldosterone were measured and echocardiography was performed with a LOGIQ S8 ultrasound device (GE Healthcare Japan Corp., Tokyo, Japan) by a specialized echocardiographer. (see Table 1 for study design).

The exclusion criteria were as follows: (1) hemodialysis; (2) hepatic dysfunction; (3) pregnancy; and (4) patients who were deemed unsuitable by the attending physician for other reasons. Adverse reactions included hypertension; tachycardia; bradycardia; renal dysfunction, which was defined as an increase in sCr levels of 50% or more; hepatic dysfunction, which was defined as an increase of 50% or more in aspartate transaminase and alanine transaminase levels; skin reactions; and allergic reactions. Adverse reactions management, such as the discontinuation of the test drug, was determined by the attending physician.

Written informed consent was obtained from all participants. The study was approved by the institutional review board and registered with the Hospital Medical Information Network (Study ID: UMIN000025419).

### 2.2. Statistical Analysis

The data are expressed as the mean ± standard error of the mean (SEM). The variables were compared by one-way analysis of variance (ANOVA). A *p* value of less than 0.05 indicated statistical significance. The statistical analysis in this study was conducted using Macintosh (MacBook Pro) with mac OS Big Sur, along with the analysis software, SPSS software (Version 28.0.0.0, IBM Inc., Chicago, IL, USA). The data were aggregated by Sekino Laboratory staff who were not involved in the study and analyzed by SATISTA (Kyoto, Japan), a company that was not involved in performing the study.

## 3. Results

A total of 60 patients were enrolled in the study (see Table 2 for the baseline characteristics). In the 12 month study period, no deaths, cardiac or renal events, thrombosis, or adverse reactions occurred. At month 12, the mean maintenance dose of CERA was 82.0 ± 4.8 μg/month. The hemoglobin levels reached 13 g/dL in 12 patients, so the CERA dose was either stopped (four patients: two at month 6 and another two at month 12) or reduced (eight patients: two at month 3, three at month 6, and another three at month 12). None of the patients who stopped taking CERA resumed intake, and the dose was not Increased in any of the patients in whom it was decreased. Two months after the initiation of CERA, all of the patients could maintain the target Hb level of between 11 and 13 g/dL with a monthly regimen.

### 3.1. Primary Endpoints

The Hb levels increased significantly from the baseline to all time points, as follows: baseline, 9.8 ± 0.1 g/dL; month 1, 11.0 ± 0.1 g/dL; month 3, 11.6 ± 0.2 g/dL; month 6, 11.7 ± 0.2 g/dL; and month 12, 12.2 ± 0.2 g/dL (all *p* < 0.001; Figure 1). Three months after the start of CERA, all of the patients could maintain the target Hb level (between 11 and 13 g/dL).

The Ht levels also increased significantly from baseline to all time points, as follows: baseline, 30.4 ± 0.4%; month 1, 33.8 ± 0.4%; month 3, 35.8 ± 0.5%; month 6, 36.4 ± 0.5%; and month 12, 37.5 ± 0.5% (all *p* < 0.001; Figure 1).

### 3.2. Secondary Endpoints

#### 3.2.1. BNP and ANP

The BNP and ANP results are shown in Figure 2. The BNP levels increased significantly from baseline to all time points, as follows: baseline, 174.2 ± 21.2 pg/mL; month 1, 121.7 ± 14.6 pg/mL; month 3, 110.1 ± 11.7 pg/mL; and month 12, 111.7 ± 12.2 pg/mL at 0 (all *p* = 0.001). The ANP levels also increased significantly from baseline to months 3, 6 and 12: baseline, 104.9 ± 9.1 pg/mL; month 3, 71.4 ± 6.3 pg/mL; month 6, 83.3 ± 7.8 pg/mL; and month 12, 83.9 ± 6.8 pg/mL (all *p* < 0.001).

#### 3.2.2. NYHA Class

The NYHA class increased significantly from baseline to months 6 and 12 (both *p* = 0.007; Table 3).

The maintenance dose of CERA was analyzed by the NYHA. The maintenance doses of CERA at 12.5 U, 25U, 50 U, 62.5 U, 75 U, 100 U, 150 U, and >150 U were NYHA at 1.5 ± 0.5, 2.0 ± 0, 1.8 ± 0.1, 2.0 ± 0, 1.9 ± 0.1, 1.7 ± 0.2, 2.0 ± 0.3, and 2.0 ± 0.2, respectively. Thus, there was no significant relationship between the maintenance dose of CERA and the NYHA, and hence no dependency effect.

#### 3.2.3. Echocardiography Measurements

The echocardiography results are shown in Table 2. There were no significant changes in the LVEF, %FS, LVEDD, LVESD, or E/e’, but the LVMI decreased significantly from baseline to month 12 (*p* = 0.008).

#### 3.2.4. BUN, sCr, eGFR, Cystatin C, and U-Alb

The results of the renal variables are shown in Table 2 and Figure 2. Compared with the baseline, the BUN levels were significantly lower at months 3 (*p* = 0.003) and 6 (*p* = 0.029). At months 3, 6, and 12, significant decreases were seen in the sCr levels (month 3, *p* < 0.001; month 6, *p* < 0.001; month 12, *p* = 0.001), eGFR (all *p* < 0.001) and the cystatin C levels (month 3, *p* = 0.03; month 6, *p* = 0.033; month 12, *p* = 0.038). There were no significant changes in the U-Alb levels over the 12 months study period.

#### 3.2.5. hs-CRP

There were no significant changes in the hs-CRP levels over the 12 months of CERA administration (Table 2).

#### 3.2.6. Ox-LDL

The Ox-LDL levels decreased significantly from baseline to months 6 (*p* = 0.036) and 12 (*p* = 0.017; Figure 2).

#### 3.2.7. Renin, Angiotensin-II, and Aldosterone

There were no significant changes in the levels of renin and aldosterone over the 12 months study period, but the angiotensin-II levels decreased significantly from baseline to month 12 (*p* = 0.003; Table 2).

## 4. Discussion

In this study, the administration of CERA improved not only anemia but also renal function, cardiac function, and oxidative stress. In addition, it was found that when the Hb levels were maintained between 11 and less than 13 g/dL, CERA was safe and did not cause any adverse reactions, such as thrombosis. Among the various ESA products, CERA was selected in this study for the following reasons: The study subjects were out-patients and visit a clinic on a monthly basis. Accordingly, a product with long-life is preferred. There is no head-to-head comparison study in heart failure between CERA and other ESA products. The half-life of CERA (4 to 6 days) is longer than that of other ESA products. A decreased dosing frequency is reported in a study that investigated switching from recombinant human Erythropoietin to CERA in dialysis patients [15]. In this study, all of the patients could successfully maintain the target Hb level of between 11 and 13 g/dL with a monthly dosing regimen, without complications.

The echocardiography showed no differences in the left ventricular function after CERA administration in many of the patients with heart failure and preserved LVEF who were in a low NYHA class. However, significant decreases were seen in the ANP, BNP, LVMI, and NYHA class after the administration of CERA, which improved the patients’ anemia and reduced the effects of cardiac stress. Another study reported that the ESA erythropoietin increased exercise tolerance, lowered the NYHA class, improved renal function, and reduced the BNP levels in patients with chronic heart failure [16]. Silverberg et al. reported that erythropoietin and an iron preparation improved the NYHA class, hospitalization rate, and diuretic requirements in patients with chronic heart failure with an LVEF of 40% or less and Hb levels of between 10 and 11.5 g/dL [17]. In a clinical study on ESA treatment in 319 patients with chronic heart failure, the ESA improved the Hb levels from 11.3 to 13.4 g/dL, but did not significantly improve the exercise tolerance, NYHA class, or quality of life; however, the exercise time was reported to be significantly longer in patients whose serum Hb levels increased by 12.0 g/dL or more [6].

ESA reduce the BNP levels, NYHA class, and hospitalization rates in patients with cardio-renal anemia syndrome, so the correction of anemia by ESA plays a role in the improvement of the clinical outcome in this subset of patients with heart failure [8]. Kuwahara et al. reported that pre-dialysis patients with Hb levels below 8.9 g/dL have lower LVEF and significantly increased LVMI and deceleration time (DcT) than patients with higher Hb levels [18]. Hayashi et al. reported that the anemia improved and LVMI decreased in nine predialysis patients with chronic renal failure who were treated with ESA; furthermore, this study also found a correlation between anemia and LVMI [19].

Similar to previous studies, the present study found that CERA had positive renal effects, such as the improvement in the sCr levels and eGFR. CERA had favorable effects on the glomerular function because it significantly reduced the levels of cystatin C, a biomarker that closely reflects glomerular function without being influenced by the patient’s diet or muscle mass. Only a few other studies have reported on the cystatin C levels of patients undergoing ESA treatment, and only one described the effects of ESA treatment on tubular injuries and showed a correlation between the cystatin C levels and decreased levels of neutrophil gelatinase-associated lipocalin, a biomarker of tubular injury, after ESA administration [20].

The effects of CERA on the renin-angiotensin-aldosterone system (RAAS) were also investigated. The study participants with chronic heart and renal failure had increased RAAS before the CERA administration. The angiotensin-II levels decreased after 12 months of CERA administration. However, these results could not be analyzed in detail due to the fact that many of the study patients were taking renin-angiotensin system inhibitors, such as mineralocorticoid receptor antagonist (MRA). It is hypothesized that CERA may not directly affect the RAAS, but that it may improve anemia by having favorable effects on the heart and kidneys, which may lead to secondary effects on the RAAS.

Oxidative stress is known to have a strong influence on the kidneys. In this study, the Ox-LDL levels decreased significantly after CERA administration. Fujiwara et al. reported that erythropoietin improved anemia and significantly decreased the oxidative stress marker 8-OHdG, carotid artery intima-media thickness, brachial-ankle pulse wave velocity, and serum asymmetrical dimethylarginine levels, and consequently slowed the progression of renal insufficiency, oxidative stress, and atherosclerosis in 15 patients with renal anemia [21]. Our results showed that CERA not only improved anemia, but also reduced renal impairment, cardiac stress, and oxidative stress, and they suggest that using CERA as a treatment for renal anemia breaks the vicious cycle of damage to the heart and kidneys by improving oxidative stress and organ derangement, thus preventing the exacerbation of both heart disease and renal disease.

In this study, an improvement of anemia by CERA was anticipated. However, it is of interest to observe the improvements in the renal functions, ANP, BNP, LVMI, and oxidative stress. The study results suggest that these are unlikely to be direct effects of CERA, but are instead due to an increase in erythropoietin by CERA. It is indicated that the production of erythropoietin is decreased in patients with heart failure [22]. Accordingly, it appears meaningful to administer ESA to patients with renal anemia. It is reported that the treatment of patients with chronic heart failure by ESA resulted in anti-oxidative effects and anti-apoptosis effects [23]. This clinical study demonstrated that Ox-LDL was decreased by CERA, which is a significant finding. During the study period, SGLT2 inhibitors were not yet approved for use in heart failure in Japan, and SGLT2 inhibitors were not popular in diabetic patients. In recent years, the sub-analysis of large-scale studies of SGLT2 inhibitors demonstrated that an increase in erythropoietin by SGLT2 inhibitor affects the LVMI, oxidative stress, and oxygen delivery, which positively impacts the cardiac and renal functions [24,25]. In this study, there was no side effect, such as cardiovascular events. An improvement of anemia by CERA and safety was demonstrated. However, the study is ongoing for 12 months of follow-up, and it is not yet known if CERA can improve the outcome of patients with heart failure through an improvement in anemia. This is one of limitations of this study. The beneficial effects of erythropoietin on the cardiovascular system have long been discussed. Erythropoietin potentially confers the cardio-protective effects via anti-apoptosis, anti-inflammation, anti-oxidation, and neovascularization [26]. However, there are still ongoing discussions about whether these effects could result in an improvement in the prognosis of heart failure. Furthermore, with regard to the potential improvement of the prognosis of heart failure by CERA, more patients should be studied for a longer follow-up period.

New drugs, including hypoxia-inducible factor prolyl-hydroxylase (HIF-PH) inhibitors, have been developed for renal anemia and are now in clinical use. These agents work by stabilizing the hypoxia-inducible factor complex and stimulating endogenous erythropoietin production, even in patients with end-stage kidney disease. Five different HIF-PH inhibitors are currently available; they are administered orally, which may be a more favorable route for patients not undergoing hemodialysis [27,28,29,30,31]. A study is currently being conducted to evaluate the effects of switching from CERA to HIF-PH inhibitors and to clarify the differences between the two types of drugs and their efficacy and potential adverse effects (Study ID: UMIN000041651). In the future, the best way to treat heart failure with renal anemia will need to be clarified.

## 5. Limitation

This was a non-controlled, single arm study at a single clinical site, and a limited number of patients were studied. In this study, priority was given to treat renal anemia in clinical practice. Accordingly, a randomized controlled clinical trial could not ethically be conducted. However, to further investigate the efficacy of CERA and its limitations, a randomized, controlled clinical trial is warranted as a future endeavor. Accordingly, the study design is not robust enough to demonstrate the efficacy of CERA. Moreover, this study is still ongoing for 12 months of follow-up. It is not known whether an improvement of anemia by CERA affects the outcome of patients with heart failure. Future studies should include more patients for a longer-term follow-up at multiple centers.

## 6. Conclusions

This study determined that CERA improves anemia and also reduces renal impairment, cardiac stress, and oxidative stress in patients with chronic heart failure and renal anemia. The result of this study is useful as a study in which CERA is switched to a new renal anemia drug, hypoxia-inducible factor prolyl-hydroxylase (HIF-PH) inhibitors.

## Figures and Tables

**Figure 1 biomedicines-11-00946-f001:**
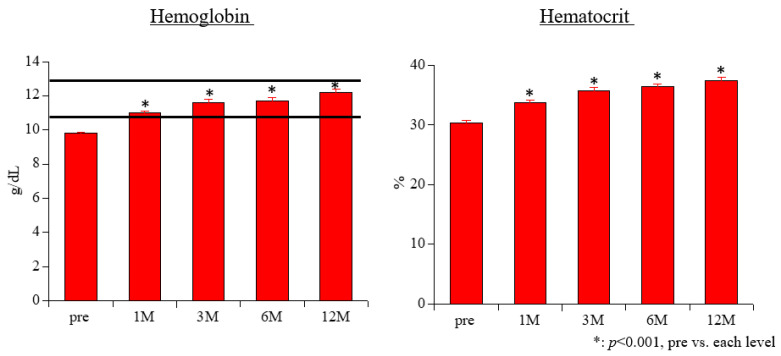
Changes in the levels of hemoglobin and hematocrit over 12 months of treatment with continuous erythropoietin receptor activator.

**Figure 2 biomedicines-11-00946-f002:**
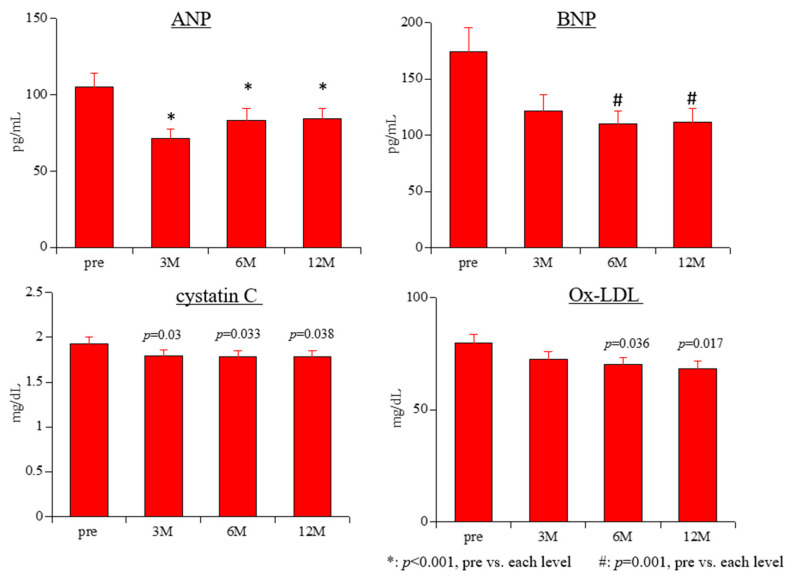
Changes in the levels of atrial natriuretic peptide, B-type natriuretic peptide, cystatin C, and oxidized low-density lipoprotein over 12 months of treatment with continuous erythropoietin receptor activator. ANP, atrial natriuretic peptide; BNP, B-type natriuretic peptide; Ox-LDL, oxidized low-density lipoprotein.

**Table 1 biomedicines-11-00946-t001:** Dosing method and the timing of each examination.

	Pre	1M	2M	3M	4M	5M	6M	7M	8M	9M	10M	11M	12M
CERA administration	50 µg	●	●	●	●	●	●	●	●	●	●	●	●
Hb, Ht	◯	◯		◯			◯						◯
ANP, BNP	◯	◯		◯			◯						◯
NYHA class	◯	◯		◯			◯						◯
Echocardiography	◯						◯						◯
BUN, Cr, eGFR	◯	◯		◯			◯						◯
Cystatin C	◯	◯		◯			◯						◯
U-alb	◯	◯		◯			◯						◯
hs-CRP	◯	◯		◯			◯						◯
Ox-LDL	◯	◯		◯			◯						◯
Renin, ang-II, ALD	◯						◯						◯

●: The dose of CERA was adjusted to maintain hemoglobin value in a range of 11 and 13 dL. The interval of intravenous dosing was monthly or every other week. ◯: measurement, CERA: continuous erythropoietin receptor activator. Hb: hemoglobin, Ht: hematocrit, ANP: atrial natriuretic peptide, BNP: B-type natriuretic peptide, NYHA: New York Heart Association, BUN: blood urea nitrogen, Cr: creatinine, eGFR: estimated glomerular filtration rate, U-alb: urinary albumin, hs-CRP: high-sensitivity C-reactive protein, Ox-LDL: oxidized low-density lipoprotein, ang-II: angiotensin-II, ALD: aldosterone.

**Table 2 biomedicines-11-00946-t002:** Patient characteristics.

Number	60
Age (years)	79.7 ± 0.85
Gender (male:female)	34:26
Etiology of heart failure	
Ischemic heart disease	25 (42%)
Valvular disease	26 (43%)
Hypertension	7 (12%)
Arrhythmia	2 (3%)
Classification of heart failure	
HFrEF	7 (12%)
HFpEF	53 (78%)
NYHA class	
I	8 (13%)
II	40 (67%)
III	12 (20%)
Risk factors	
Diabetes mellitus	26 (43%)
Hypertension	44 (73%)
Dyslipidemia	34 (57%)
Hyperuricemia	38 (63%)
Obesity	10 (17%)
Smoking	15 (25%)
Oral medication	
Calcium antagonist	24 (40%)
ACE- inhibitor	32 (53%)
ARB	12 (20%)
Renin- inhibitor	3 (5%)
Aldosterone blocker	28 (47%)
Beta blocker	49 (82%)
Furosemide	47 (78%)
Tolvaptan	10 (17%)

**Table 3 biomedicines-11-00946-t003:** Changes of each blood and urine test and echocardiographic data.

	Pre	3 Months	*p* Value	6 Months	*p* Value	12 Months	*p* Value
NYHA class	2.07 ± 0.08	2.00 ± 0.08	n.s.	1.90 ± 0.70	0.007	1.90 ± 0.70	0.007
BUN (mg/dL)	32.2 ± 1.4	27.7 ± 1.2	0.003	28.3 ± 1.4	0.029	29.3 ± 2.0	n.s.
Serum creatinine (mg/dL)	1.70 ± 0.06	1.48 ± 0.58	<0.001	1.49 ± 0.67	<0.001	1.47 ± 0.74	0.001
eGFR (mL/min/1.73 m^2^)	29.8 ± 1.17	34.9 ± 1.41	<0.001	34.2 ± 1.51	<0.001	35.6 ± 1.72	<0.001
Urinary albumin (mg/g·Cr)	381.9 ± 115.9	398.7 ± 143.4	n.s.	299.4 ± 81.2	n.s.	267.0 ± 95.0	n.s.
Renin (ng/mL/h)	7.95 ± 1.66	-	-	9.47 ± 2.08	n.s.	9.94 ± 2.68	n.s.
Angiotensin-II (pg/mL)	25.2 ± 3.9	-	-	24.3 ± 6.5	n.s.	12.9 ± 1.52	0.003
Aldosterone (pg/mL)	146.4 ± 19.0	-	-	136.9 ± 17.0	n.s.	130.2 ± 13.3	n.s.
hs-CRP (mg/dL)	0.31 ± 0.06	0.23 ± 0.04	n.s.	0.27 ± 0.07	n.s.	0.35 ± 0.10	n.s.
Ejection fraction (%)	59.8 ± 1.6	-	-	60.7 ± 1.4	n.s.	60.8 ± 1.3	n.s.
Fractional shortening (%)	32.0 ± 1.0	-	-	32.7 ± 0.9	n.s.	32.8 ± 0.9	n.s.
LVDd (mm)	46.9 ± 0.9	-	-	46.2 ± 1.1	n.s.	45.4 ± 1.0	n.s.
LVDs (mm)	31.5 ± 1.0	-	-	31.3 ± 1.0	n.s.	30.3 ± 1.0	n.s.
LVMI	174.5 ± 7.8	-	-	161.6 ± 6.9	n.s.	153.6 ± 6.4	n.s.
E/e’ratio	13.3 ± 0.9	-	-	12.8 ± 0.9	n.s.	14.0 ± 1.1	n.s.

NYHA class: New York Heart Association functional classification, n.s.: not significant, BUN: Blood urea nitrogen, eGFR: estimated glomerular filtration rate, hs-CRP: high-sensitivity C-reactive protein, LVDd: left ventricular end-diastolic dimension n, LVDs: left ventricular end-systolic dimension, LVMI: left ventricular mass index.

## Data Availability

We have not asked the participants in this study whether the data can be shared publicly, so the supporting data are not available.

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
