# Peer review of "A Single Arm Clinical Study on the Effects of Continuous Erythropoietin Receptor Activator Treatment in Non-Dialysis Patients with Chronic Heart Failure and Renal Anemia"

_biomedicines, 2023, doi:10.3390/biomedicines11030946_

Round 1
Reviewer 1 Report
The paper "A clinical study on the effects of continuous erythropoietin receptor activator treatment in non-dialysis patients with chronic heart failure and renal anemia" presents the effect of continuous erythropoietin receptor activator on the cardiac function of 60 patients. The results show that anemia improved and renal impairment, cardiac stress, along with oxidative stress were reduced.
- Originality/Novelty: The work is original. The proposed solution can be further used in healthcare research.
- Significance: The results of the research are interpreted properly. The conclusions are justified and supported by the results which are displayed in tables, as well as graphically, along with descriptions and comparisons, but they need to be enriched. All hypotheses were specified and the outcomes were carefully analysed.
- Quality of Presentation: The article is written appropriately, respecting the logical succession of sections. Data and analyses are presented graphically and inside tables.
Please provide the technical characteristics of the PC on which the analysis was done.
Please avoid the usage of the pronoun "we". Reformulate all such phrases in order to have a more formal approach.
The conclusion section contains just one phrase. Please enrich the conclusions section and mention any future works which will be investigated.
The reference section must include more recent papers from the last 3 years.
Section 3, Results, should not start with "Patients:". It should just be mentioned that the study was done on 60 people who enrolled in the trial.
Figure 1 is not properly described. The Hb levels are enumerated, but the values are not explained if they belong to the right range or not. What is the significance? The same is valid for secondary endpoints.
Please further explain the values from Table 2.
- Scientific Soundness: The study of the proposed continuous erythropoietin receptor activator is well documented and will be useful for further research.
- Interest to the Readers: The conclusions will surely interest the readers of the Biomedicines journal, and not only them, as the healthcare domain is of great interest.
- Overall Merit: The benefit of publishing this paper consists in a good documented state-of-the-art article regarding the effects of continuous erythropoietin receptor activator.
- English Level: The level of English language is advanced. Through the entire paper, the language was appropriate and understandable, being easy to follow the flow from the beginning.
Author Response
Dear Reviewer 1
Thank you for your detailed review of our manuscript and the instructions for revisions. The manuscript has been revised in order to improve the quality of the English by a native English-speaking medical doctor.
Our article is now ready for re-submission with the revisions detailed below and we consider that our report now has a stronger impact.
We would appreciate it if you could assess the revised manuscript.
- Please provide the technical characteristics of the PC on which the analysis was done.
→Statistical analysis of this study was conducted using Macintosh (MacBook Pro) with mac OS Big Sur, along with the analysis software, SPSS software (Version 28.0.0.0, IBM Inc, IL, USA).
- Please avoid the usage of the pronoun "we". Reformulate all such phrases in order to have a more formal approach.
→Thank you for your valuable suggestion. We revised the manuscript accordingly.
- The conclusion section contains just one phrase. Please enrich the conclusions section and mention any future works which will be investigated.
→Thank you for your valuable comment. We also included potential future studies in Conclusion section.
- The reference section must include more recent papers from the last 3 years.
→Thank you for your valuable suggestion. Latest references were added.
- Section 3, Results, should not start with "Patients:". It should just be mentioned that the study was done on 60 people who enrolled in the trial.
→Thank you for your valuable comment. We revised the manuscript accordingly.
- Figure 1 is not properly described. The Hb levels are enumerated, but the values are not explained if they belong to the right range or not. What is the significance? The same is valid for secondary endpoints.
→Thank you for your valuable comment. P value was wrong. We corrected it.
- Please further explain the values from Table 2.
→Thank you for your valuable comment. P value was wrong. We corrected it.

Reviewer 2 Report
This is a clinical study which sought to determine the effect of continuous erythropoietin receptor activator (CERA) on cardiac and renal function, neurohumoral factors, inflammatory and oxidative stress markers for heart failure concomitant with renal anemia. The Reviewer assumes that this study was conducted as a single-arm intervention study. After 1-year CERA administration significant improvement of symptom (NYHA), Hb, renal function (creatinine, eGFR, cystatin C) and LVMI were observed. From these findings the Authors concluded that CERA was effective in ameliorating the pathology of cardio-renal-anemia syndrome.
The findings look reasonable. But there are numerous issues in the presentation, discussion, and English in this manuscript.
Major comments:
1. Improvement of Hb by CERA is not surprising. The matter of interest is improvement of the other indices such as renal function, LVMI and oxidative stress. Are these a direct or indirect effect by CERA? The Authors need to discuss more in detail.
2. Lines 62-63. Even though the Authors stated that only a few clinical studies evaluated the influence of CERA on the heart, the difference between CERA and the other ESA agents were not discussed at all. What is the distinctive feature of CERA? The Authors can compare this study finding and the other studies which used the other ESA agents.
3. Lines 50-52. The mechanism of anemia in patients with heart failure might be complex and multifactorial, not only decreased production of erythropoietin. As the Authors stated, cardio-renal anemia syndrome forms vicious cycle which leads to poor prognosis. Further, this pathological condition might badly influence on the response to certain heart failure treatment (Circ J 2018; 82: 691 – 698). Since this point is fundamental in this paper, the Authors need to describe more detail. Also, these descriptions need reference.
4. Subsection 2.1 Study design is obscure. The Reviewer assumes that this is a single-arm intervention study. Does this study include the patients with acute decompensated heart failure (ADHF) or stable chronic HF? Outpatients or inpatients or both? Was “standard treatment of heart failure” (line 73) required for study inclusion?
5. Line 72. What is “a history of heart failure”? Do the Authors mean the episode of ADHF, (since this study included patients with NYHA I)?
6. Lines 123-131. The time course of CERA dosing is too complex to follow. Figure or Table depicting this might be helpful.
7. How about the dose – effect relationship between CERA and the outcomes?
8. Limitation section is missing. All studies have limitations with no exception. For example, single arm design which does not have control group is an important limitation.
9. There are significant issues in English expression and grammar. The Reviewer strongly recommend this manuscript. Several examples (not limited to these) are shown below.
- Abstract, lines 63, 65 “research studies”
- Line 39-40. This sentence should be reorganized.
- Line 99 (etc.). After “initiation of ” ESA administration
- Line 124 “cridoac”
- Figure 2. “*: p<0.05 vs. pre” might be better.
- Line 204. “by”
- Lines 244-247. These 2 sentences should be reorganized.
10. Lines 187-188. The Authors stated “we found that CERA could be used safely without adverse reactions, such as thrombosis”. It should be stated in the Results section.
Minor comments:
Lines 75-78. The description on iron deficiency anemia is conflicting. Please re-construct the sentences.
How many patients received SGLT-2 inhibitors at baseline?
“Aldosterone blocker” should be amended to “mineralocorticoid receptor antagonist (MRA)”
Author Response
Dear Reviewer 2
Thank you for your detailed review of our manuscript and the instructions for revisions. The manuscript has been revised in order to improve the quality of the English by a native English-speaking medical doctor.
Our article is now ready for re-submission with the revisions detailed below and we consider that our report now has a stronger impact.
We would appreciate it if you could assess the revised manuscript.
Major comments:
- Improvement of Hb by CERA is not surprising. The matter of interest is improvement of the other indices such as renal function, LVMI and oxidative stress. Are these a direct or indirect effect by CERA? The Authors need to discuss more in detail.
→Thank you for valuable comment. The study result is unlikely to be due to direct effect of CERA. It is most likely that an improvement in anemia resulted in better outcome. In addition, CERA certainly increases erythropoietin level, and this study clearly demonstrated it. On this point, we added a reference and included a comment in Discussion section.
- Lines 62-63. Even though the Authors stated that only a few clinical studies evaluated the influence of CERA on the heart, the difference between CERA and the other ESA agents were not discussed at all. What is the distinctive feature of CERA? The Authors can compare this study finding and the other studies which used the other ESA agents.
→Thank you very much for very valuable comment. The characteristics of CERA is a long half-life. There is not head-to-heard comparison study between CERA other ESA product in patients with heart failure. However, there is a study in dialysis patients. we added a reference and included a comment in Discussion section.
- Lines 50-52. The mechanism of anemia in patients with heart failure might be complex and multifactorial, not only decreased production of erythropoietin. As the Authors stated, cardio-renal anemia syndrome forms vicious cycle which leads to poor prognosis. Further, this pathological condition might badly influence on the response to certain heart failure treatment (Circ J 2018; 82: 691 – 698). Since this point is fundamental in this paper, the Authors need to describe more detail. Also, these descriptions need reference.
→Thank you very much for very valuable comment. We added a recommended article in Reference and also included a comment in Introduction section.
- Subsection 2.1 Study design is obscure. The Reviewer assumes that this is a single-arm intervention study. Does this study include the patients with acute decompensated heart failure (ADHF) or stable chronic HF? Outpatients or inpatients or both? Was “standard treatment of heart failure” (line 73) required for study inclusion?
→This is the most important point in clinical investigation, and we appreciate your comment. We revised Study design section to make it clearer.
- Line 72. What is “a history of heart failure”? Do the Authors mean the episode of ADHF, (since this study included patients with NYHA I)?
→This is the most important point in clinical investigation, and we appreciate your comment. We revised Study design section to make it clearer.
- Lines 123-131. The time course of CERA dosing is too complex to follow. Figure or Table depicting this might be helpful.
→We thank you for your sound comment. In this study, CERA was initially administered either once every other week or monthly so as to adjust Hb in a range of 11 to 13 g/dL. However, all patients could maintain such a level on a monthly regimen after two-month treatments. This was described in Result section.
- How about the dose – effect relationship between CERA and the outcomes?
→Thank you very much for very valuable comment. We considered CERA and the CERA dose. Dose-dependent effects could not be observed. This was discussed in Result section.
- Limitation section is missing. All studies have limitations with no exception. For example, single arm design which does not have control group is an important limitation.
→Your comment is correct. This is the most important point in clinical investigation. This study is associated with Limitations. We created Limitation section and made comments.
- There are significant issue
s in English expression and grammar. The Reviewer strongly recommend this manuscript. Several examples (not limited to these) are shown below.
- Abstract, lines 63, 65 “research studies”
- Line 39-40. This sentence should be reorganized.
- Line 99 (etc.). After “initiation of ” ESA administration
- Line 124 “cridoac”
- Figure 2. “*: p<0.05 vs. pre” might be better.
- Line 204. “by”
- Lines 244-247. These 2 sentences should be reorganized.
→We thank you for a comment to details. We corrected the manuscript per your comments.
- Lines 187-188. The Authors stated “we found that CERA could be used safely without adverse reactions, such as thrombosis”. It should be stated in the Results section.
→We thank you for your comment. We added it in Result section.
Minor comments:
Lines 75-78. The description on iron deficiency anemia is conflicting. Please re-construct the sentences.
→We thank you for your valuable comment. It was indeed an error. We corrected it.
How many patients received SGLT-2 inhibitors at baseline?
→We thank you for your valuable comment. During the study period, SGLT2 inhibitors were yet approved for use in heart failure in Japan, and SGLT2 inhibitors were not popular in diabetic patients neither. We added this point in Discussion section.
“Aldosterone blocker” should be amended to “mineralocorticoid receptor antagonist (MRA)”
→We thank you for your comment and corrected it accordingly.

Reviewer 3 Report
The article analyze the administration of CERA, and this drug appears to improve anemia and also reduces renal impairment, cardiac stress, and oxidative stress in patients with chronic heart failure and renal anemia. However, as the study is prospective clinical research, the patient selection criteria must be very clear. Thus, the authors need to include the inclusion criteria, where the description of the study population should be much improved.
The introduction and the conclusions are also very poor, it does not say why the analysis of those biomarkers, . the results are very few and the novelty is also not properly explained.
Author Response
Dear Reviewer 3
Thank you for your detailed review of our manuscript and the instructions for revisions The manuscript has been revised in order to improve the quality of the English by a native English-speaking medical doctor.
Our article is now ready for re-submission with the revisions detailed below and we consider that our report now has a stronger impact.
We would appreciate it if you could assess the revised manuscript.
The article analyze the administration of CERA, and this drug appears to improve anemia and also reduces renal impairment, cardiac stress, and oxidative stress in patients with chronic heart failure and renal anemia. However, as the study is prospective clinical research, the patient selection criteria must be very clear. Thus, the authors need to include the inclusion criteria, where the description of the study population should be much improved.
→We thank you for valuable comment. The target patient population in Study protocol was revised.
The introduction and the conclusions are also very poor, it does not say why the analysis of those biomarkers, the results are very few and the novelty is also not properly explained.
→We thank you for your detailed comments. Introduction, Conclusions and Discussions were revised, and additional remarks were included.

Round 2
Reviewer 2 Report
The Authors made some correction according to the Reviewer’s comments. However, the majority of the Authors’ response was insufficient and far from satisfaction. Further, the Authors did not specify where the amendment is in the manuscript.
Major comments:
1. Response to Comment #1. The Authors stated an improvement in anemia resulted in better outcome. The Reviewer does not agree with the Authors. There have been long debate on the cardiovascular beneficial effect of erythropoietin. EPO plays a role in cardioprotection potentially through its anti-apoptotic, anti-inflammatory, anti-oxidant and angiogenic action (Curr Pharm Des . 2011;17(15):1517-29).
2. Response to Comment #2. The Reviewer did not intend head-to-head comparison between CERA and conventional EPO agents. If there is no distinctive feature of CERA which potentially provide clinical benefit, it is not worth testing it.
3. Response to Comment #3. The Reviewer cannot find any additional comments or discussion on the complex pathophysiology of anemia seen in heart failure..
4. Response to Comment #4. The Reviewer does not feel that the Authors responded to the Reviewer’s comments properly. Study design is not specified and still obscure. There is no “Study design” section.
5. Response to Comment #6. The Reviewer does not feel that the Authors responded to the Reviewer’s comments properly.
6. Response to Comment #8. Limitation is poorly described. The Authors did not mention even what the Reviewer’s stated.
Author Response
Dear Reviewer,
I thank you for your careful review again. Please find below amendments per your comments.
Your critical review is highly appreciated.
- Response to Comment #1. The Authors stated an improvement in anemia resulted in better outcome. The Reviewer does not agree with the Authors. There have been long debate on the cardiovascular beneficial effect of erythropoietin. EPO plays a role in cardio-protection potentially through its anti-apoptotic, anti-inflammatory, anti-oxidant and angiogenic action (Curr Pharm Des. 2011;17(15):1517-29).
→This is a very important comment. I agree with you that there are some room for discussions if an improvement of anemia results in better outcome in patients with heart failure. There are several reports of cardio-protective effects of EPO, as you pointed out. This study is not the direct evidence that an improvement of anemia leads to better outcome. We will discuss this matter in Discussions and added the reference that you cited.
- Response to Comment #2. The Reviewer did not intend head-to-head comparison between CERA and conventional EPO agents. If there is no distinctive feature of CERA which potentially provide clinical benefit, it is not worth testing it.
→I thank you for your comment. It is reported that conventional ESA agents is cardio-protective. The most important advantage of CERA is longer half-life, and its efficacy in patients with renal anemia. However, there is no clinical report in patients with heart failure who received CERA. Therefore, we considered that this study was warranted. We commented this in Introduction section.
- Response to Comment #3. The Reviewer cannot find any additional comments or discussion on the complex pathophysiology of anemia seen in heart failure.
→Although we provided an explanation in Introduction section, it may not be sufficient. We have added a reference and revised Introduction section.
- Response to Comment #4. The Reviewer does not feel that the Authors responded to the Reviewer’s comments properly. Study design is not specified and still obscure. There is no “Study design” section.
→My apology for failure to your question. We revised the manuscript with plain language.
- Response to Comment #6. The Reviewer does not feel that the Authors responded to the Reviewer’s comments properly.
→We are very sorry for this. We prepared a Table to improve readability. Route of administration was intravenous. It was administered monthly or every other week. We maintained hemoglobin values in a range of 11 to 13 g/dL.
- Response to Comment #8. Limitation is poorly described. The Authors did not mention even what the Reviewer’s stated.
→This is a very important comment. There are several Limitations in this study. One of them is the fact that we could not demonstrate evidence that CERA could improve the outcome. We have added this.

Reviewer 3 Report
The article has been improved, and the suggestions have been implemented.
Author Response
We are most grateful to you for the helpful comments about our manuscript.
We hope that the revised manuscript is now acceptable for publication.

Round 3
Reviewer 2 Report
Response to Comment #2. It looks like the Authors did not catch what the Reviewer meant. What is the difference between the present study and the previous studies which tested the other EPO in HF patients ?
Response to Comment #5 in 2nd round. The Reviewer does not feel that the Authors responded to the Reviewer’s comments properly. The Reviewer could not find new Table.
Author Response
We are most grateful to you for the helpful comments about our manuscript.
Although Table 1 was submitted, it was not reflected in the manuscript. My apology for this inconvenience. Table 1 was added.
